# UWB Localization with Battery-Powered Wireless Backbone for Drone-Based Inventory Management

**DOI:** 10.3390/s19030467

**Published:** 2019-01-23

**Authors:** Nicola Macoir, Jan Bauwens, Bart Jooris, Ben Van Herbruggen, Jen Rossey, Jeroen Hoebeke, Eli De Poorter

**Affiliations:** IMEC, IDLab, Department of Information Technology, Ghent University, 9000 Ghent, Belgium; jan.bauwens2@ugent.be (J.B.); bart.jooris@ugent.be (B.J.); ben.vanherbruggen@ugent.be (B.V.H.); jen.rossey@ugent.be (J.R.); jeroen.hoebeke@ugent.be (J.H.)

**Keywords:** Ultra-Wideband (UWB), drone inventory, low-energy, easy installation, infrastructure-light

## Abstract

Current inventory-taking methods (counting stocks and checking correct placements) in large vertical warehouses are mostly manual, resulting in (i) large personnel costs, (ii) human errors and (iii) incidents due to working at large heights. To remedy this, the use of autonomous indoor drones has been proposed. However, these drones require accurate localization solutions that are easy to (temporarily) install at low costs in large warehouses. To this end, we designed a Ultra-Wideband (UWB) solution that uses infrastructure anchor nodes that do not require any wired backbone and can be battery powered. The resulting system has a theoretical update rate of up to 2892 Hz (assuming no hardware dependent delays). Moreover, the anchor nodes have an average current consumption of only 27 mA (compared to 130 mA of traditional UWB infrastructure nodes). Finally, the system has been experimentally validated and is available as open-source software.

## 1. Introduction

The constantly increasing success of e-commerce requires new supply chain solutions at every stage of operations, including (continuous) inventory management [1]. Industry 4.0 digitization (Also known as the next industrial (or data) revolution: using smart technologies in industry to collect data ) with goods being produced more flexibly, impacts inventory management. Also for rental companies, the trend towards “X-as-a-service” (XaaS) (XaaS refers to the growing diversity of services available over the Internet via cloud computing as opposed to being provided locally, or on premises.) requires a continuous stock management. To realize this flexibility, powerful warehouse management software [2] has been introduced. On the warehouse floor, we have RFID or barcode based solutions and even robots with RFID readers. However, although these warehouse management systems provide some automation, most traditional inventory checking approaches remain very labor-intensive and time-consuming in vertical warehouses, involving reach trucks, forklifts, man-cages or scissor-lifts and taking up hundreds of man-hours every counting cycle. In addition, human errors occur frequently, thereby impacting the quality of the data and resulting in additional lost man-hours. Finally, as the racks can be stacked quite high (up to 16 m), manual inventory tracking can result in serious accidents.

To overcome these limitations, automated inventory management solutions are needed, especially in larger-sized warehouses with high racks. For this purpose, recent years have witnessed the surge of research interest on micro unmanned aerial vehicles (UAVs) for stock counting and localization [3,4,5]. As the logical heir of ground based mobile robots, UAVs have the aerial manoeuvring ability to easily avoid obstacles and provide an excellent bird’s-eye view, which is beneficial for mapping and monitoring tasks.

In an outdoor environment, space-based satellite navigation systems such as Global Positioning System (GPS) are able to provide an absolute position for the UAV with an approximate error of 2–5 m in civilian applications, allowing automated outdoor missions. Since GPS units are unable to work reliable in an indoor environment, this approach is not feasible indoors. Moreover, a more precise positioning accuracy is required to fly between narrow racks, thereby achieving autonomous navigation of micro UAVs. Instead, such systems can replace the GPS by anchor-based localization systems. Typically, anchors are placed at points with known coordinates, and the distance from an anchor to the moving UAV can be estimated from RF signals for the localization purpose. Different kinds of ranging techniques have been proposed for the distance estimation: Time of Flight (ToF), Time of Arrival (ToA) and Received Signal Strength Indication (RSSI).

One emerging RF technology for accurate localization is Ultra-Wideband (UWB) [6] ranging technology, which is robust to multipath and non-line-of-sight (NLOS) effects [7], and can achieve a cm-level ranging error [8]. The UWB radio modules can estimate inter-module distance by measuring the transmission and reception time of UWB pulses. In contrast with other RF systems, UWB pulses can be transmitted between 3 GHz and 5 GHz and have an RF bandwidth of 500 MHz or 1.4 GHz. The increased bandwidth not only avoids the interference with other types of RF signals, e.g., remote control and WiFi signals. Moreover, the ultra-short duration pulses permit an easy filtering method to deal with the multipath effect, and provide an accurate timestamp allowing determination of ToF.

In large warehouses of multiple square kilometers, such cabling and power infrastructure costs are prohibitive, limiting the adoption of large-scale UWB systems. Most UWB solutions from scientific literature are not well-suited due to one or more of the following reasons.
A large number of scientific papers focus on improving accuracies and assume small scale set-ups that do not scale cost-efficiently towards large warehouses [9,10,11].Many proposed UWB localization solutions require Ethernet or pre-existing WiFi access points to provide connectivity between anchor nodes [12,13].Always-on UWB radios consume significantly more energy than traditional radios, thereby limiting the possibility of battery or energy-harvesting-powered anchor nodes.

To remedy this, we propose a MAC protocol for an UWB localization system using battery-powered anchors with a long battery lifetime and which can be easy installed (e.g., plug and play) without requiring wired communication infrastructure.

The main contributions of this paper are the following.
Design of a scalable localization system that requires minimal pre-existing infrastructure by combining two wireless technologies: sub-GHz for IoT-standardized long-range wireless communication backbone and UWB for localization.Optimization of the energy consumption of the UWB radio by designing a multi-technology, duty cycling time-slotted UWB MAC protocol.Theoretical evaluation of the design choices on overall system performance in terms of update rate, energy consumption, range and scalabilityExperimental validation of the system for two real-life scenarios: autonomous drone navigation and tracking of runners in sport halls.Providing the source-code of the UWB hardware and MAC protocol software as open-source contributions [14].

The remainder of this paper is organized as follows. First, Section 2 describes related work regarding drone based inventory-taking and infrastructure-light, easy-installable UWB solutions. Section 3 describes the design of our UWB localization system. Next, Section 4 describes the implementation of the system. Section 5 theoretically and experimentally evaluates the overall performance. Finally, Section 6 mentions future work and Section 7 concludes the paper.

## 2. Related Work

### 2.1. Drone-Based Inventory Taking

Although the use of drones for inventory taking in warehouses has been hailed as a major commercial breakthrough, existing commercial and academic solutions are still in their infancy. Commercial solutions such as DroneScan [15] rely on human drone operators, thereby still requiring expensive manual interventions. In contrast, fully autonomous solutions such as introduced in [3,4,5] use UWB for autonomous drone navigation, but have been demonstrated only in small areas with cabled infrastructure. Finally, drone based inventory-taking solutions such as proposed in [16] use highly accurate but also very expensive motion capturing cameras to localize the drone. As such, to a large extent, the vision of autonomous drone inventory taking is hindered due to the lack of affordable, scalable and easy to install localization solutions that can be installed in large warehouses without requiring significant infrastructure investments such as new cabling.

### 2.2. Infrastructure-Light UWB Solutions

This lack of easily installable UWB infrastructure is confirmed when looking at commercial and academic localization solutions. Although most academic research papers focus on small-scale localization setups, a small set of academic and commercial solutions exist that work at larger scale. Table 1 gives an overview of several of these solutions that provide coverage over large areas. Based on this Table, it is clear that most available solutions require either an Ethernet or WiFi infrastructure which covers the complete area and DC power or PoE (Power over Ethernet).

## 3. UWB Localization System Design

### 3.1. High Level Design

Our solution aims to minimize two shortcoming in currently available UWB solutions, namely, (i) the need for a wired backbone and (ii) the large energy consumption of the anchor nodes.
To remedy the need for wired access points, some UWB solutions [12] have proposed the use of WiFi based multi-hop mesh networks. However, a multi-hop wireless networking approach has several disadvantages. (i) Multi-hop networking protocols are scalable towards only a limited number of hops and are thus not suitable for very large warehouse environments, (ii) data transmissions required multiple hops thereby reducing the available throughput and impacting reliability and (iii) WiFi cards typically have high energy consumption, thus limiting their suitability for battery powered anchor nodes. Instead, our solution connects UWB anchor nodes using a backbone of low-power sub-GHz communications which have a much larger range, of up to several kms [21]. In addition, these long ranges can even be extended using RPL (IPv6 Routing Protocol for Low-Power and Lossy Networks) [22], which has been shown to be more scalable than WiFi ad-hoc routing protocols, with network scaling up to hundred of devices [23,24].To reduce the anchor node energy consumption, we observe that most UWB positioning systems use always-on UWB radios, which are highly energy consuming. Since drones will not be present continuously at each location, we propose to turn off the UWB radio when no drones are nearby. Instead, UWB anchor nodes continuously listen using their low power sub-GHz radio. The drone will activate anchor nodes using an low-energy sub-GHz activation beacon when he is nearby, but only (i) when he actually wants to range with the anchor node and (ii) during a small TDMA based time slot. As such, anchor nodes remain in a low-power mode most of the time, and only activate the UWB radio during short timeslots when drones are nearby.

In Figure 1, our proposed solution is summarized using a high-level system diagram.

### 3.2. MAC Design

The MAC protocol uses a TDMA channel access method, which partitions time in repeating superframes. A superframe is a sequence of slots, where each slot corresponds to a specific action to take (e.g., receiving or transmitting a UWB message). This requires all participating devices to be synchronized. Since UWB radios are more energy-hungry than common narrowband IoT transceivers, a separate transceiver is used for all communication that is not related to ranging such as synchronization and reporting. 

(i) Synchronization is initiated by the drone during the first slot of the superframe where the mobile tag transmits a sub-GHz beacon message which is received by all nearby anchor nodes (’Beacon’ slot in Figure 2). The main advantage of choosing the tag as beacon transmitter is that no multi-hop synchronization is required, because only the in-range anchors of the mobile tag need to be synchronized. Furthermore, the beacon contains a list of anchors with which the mobile tag wants to range, as well as their assigned ranging TDMA slot. All anchor nodes receiving the beacon know if and when they have to activate their UWB radio. In case the anchor node address is not included in the list, the anchor node can go to sleep mode until the next superframe.

(ii) Next, a single UWB polling message is first broadcasted (’Poll’ slot in Figure 2) to all nearby anchor nodes that turned on their UWB radio (The ranging scheme is discussed in Section 3.2.1.)

(iii) Afterwards, UWB slots are provided for sequentially calculating the distance to all selected anchor nodes (’Ranging slot’ in Figure 2).

(iv) After each ranging slot, a reporting slot is provided to send the calculated distance to the drone and/or a back-end system. Border routers are continuously listening for reporting messages and forward them to a RTLS (Real-time Localization system) to calculate the current position. In contrast to beacon messages, which are used for local anchor node selection and TDMA synchronization, reporting messages should be transmitted over a long distance. As such, the sub-GHz radio is configured for long range communications (For example, the IEEE 802.15.4 CC1200 868 MHz receiver can transmit over a large distance (varying from 292 m to 1902 m depending on the bitrate)). 

The complete structure of the superframe and an overview of the transmitted messages are illustrated in Figure 2.

#### 3.2.1. Ranging Scheme

To calculate the range (distance) to each anchor node, a two-way message exchange approach is used (The two-way-ranging approach avoids the need of ns-level synchronization between the anchor nodes, which is difficult to obtain in large warehouses with no GPS reception and no pre-existing synchronization infrastructure.) In Equation (Equation 1) the propagation time is calculated by subtracting the reply time from the total round-trip time (RTT) and dividing by two.
(1)$$\tau_{prop}=\frac{1}{2}\left(\tau_{round}-\tau_{reply}\right)$$

However, cm-level accurate positioning requires sub-nanosecond time of arrival errors [25]. The TWR (two-way-ranging) approach comes with number of sources of error due to clock drift and frequency drift. The clock drift between two ranging devices can be expressed as the difference of the fixed frequency error $\epsilon_{A},\epsilon_{B}$ with respect to the nominal oscillator frequency. The dominant error in the ranging accuracy of this scheme is strong dependent on $\tau_{reply}$ and is given in Equation (Equation 2). Assuming a reply time $\tau_{reply}$ of 1 ms, we end up with an error of up to 4 ns, which corresponds with a ranging error of 1.2 m. In Figure 3a, a plot of the ranging error in function of the frequency error is shown.
(2)$$\tau_{error}=\frac{1}{2}\tau_{reply}\left(\epsilon_{A}-\epsilon_{B}\right)$$

To remedy this, the symmetric double-sided two-way-ranging (SDS-TWR) [26] scheme uses a three-way message exchanges as shown in Figure 4. Because an extra message is sent, two round-trips are performed. And because two reply times are being used in the calculation, the local clock drift errors $\epsilon_{A}$ and $\epsilon_{B}$ are now eliminated. The error in the ranging accuracy is now dependent upon on the difference between the two reply times. As a result, the error in the ranging accuracy is much smaller as plotted in Figure 3b. Therefore, we restrain $\tau_{reply1}$ and $\tau_{reply2}$ to be equal (or as close to equal as possible). In Equation (Equation 3) the propagation time is calculated and in Equation (4) the error is calculated.

(3)$$\begin{array}{ccc} \tau_{prop} & = & \frac{1}{4}\left(\tau_{round1}-\tau_{reply1}+\tau_{round2}-\tau_{reply2}\right) \end{array}$$

(4)$$\begin{array}{ccc} \tau_{error} & = & \frac{1}{4}\Delta_{reply}\left(\epsilon_{A}-\epsilon_{B}\right) \end{array}$$

To allow flexible ranging schemes where achieving equal reply times can be challenging, it is recommended to use asymmetric double-sided TWR [25]. In this scheme the constraint of both reply times to be equal is released. The propagation time is calculated by using Equation (Equation 5).
(5)$$\tau_{prop}=\frac{\tau_{round1}\tau_{round2}-\tau_{reply1}\tau_{reply2}}{\tau_{round1}+\tau_{round2}+\tau_{reply1}+\tau_{reply2}}$$

The typical clock induced error is in the low picosecond range even with 20 ppm crystals. At these error levels the precision of determining the arrival time of the messages at each of the receivers is a more significant contributor to overall propagation time error than the clock-induced error. Where the clock in device A runs at $k_{a}$ times the desired frequency and the clock in the device B runs at $k_{b}$ times the desired frequency and both $k_{a}$ and $k_{b}$ are close to 1. Even with a relatively large UWB operating range of say 100 m, which corresponds to a propagation time of 333 ns, the error is 6.7 picoseconds which is approximately 2.2 mm.
(6)$$\tau_{error}=\tau_{prop}\left(1-\frac{k_{a}+k_{b}}{2}\right)$$

When ranging with multiple anchors, a standard approach is to do a three way message exchange to each anchor. For *n* anchors, this results in 3n messages. This number of exchanged messages can be decreases when customized schemes are used. For instance, the first message transmitted (poll-message) can be sent only once to all the anchors, after which each anchor responds sequentially. This increases the ranging update frequency and decreases the number of messages exchanged to 1+2n without any loss in accuracy because of the asynchronous double sided TWR formula is used. This is the approach taken in our solution, and is the reason why a single UWB poll message is transmitted to all anchor nodes, followed by two UWB messages per anchor node (’Response’ and ’Final’ messages), see Figure 2.

### 3.3. Collision Avoidance and Multiple Mobile Tags

One of the major drawbacks to the use of UWB radios is their limited capability to detect the presence of other UWB transmissions. This lack of carrier sensing results in increased collisions and high packet loss when multiple mobile tags need to be localized [27]. However, our approach uses a sub-GHz beacon to synchronize the nearby anchor nodes to the mobile tag TDMA scheme prior to any UWB transmissions. As such, this sub-GHz beacon can be used as ’virtual’ carrier sense by other mobile tags. Mobile tags overhear the synchronization beacon from other nodes, and hence know to defer the start of their own superframe until after the other superframe is finished, thereby avoiding UWB packet collisions. As such, superframes from multiple mobile drones will not overlap.

### 3.4. Anchor Node Selection

At the start of each superframe, the tag selects a set of anchors which are woken up to range with. This process consists of two phases: (i) bootstrapping and (ii) anchor node selection during operation lifetime.
During *bootstrapping*, the UAV has no prior information regarding its position. As such, during initialization, the set of neighbors is populated with random anchor node IDs, selecting anchor nodes at random until the UAV receives at least one ranging message.During the *operational lifetime*, the set is continuously updated to use the anchor nodes that are nearest. To this end, at the end of each superframe, the UAV selects the most nearby anchor that the tag is currently ranging with. For subsequent superframes, the UAV each time selects the set of anchor nodes that are closest to the nearest anchor node (prior to deployment, the drone is provided either with all anchor node locations, or with a dictionary where for each anchor node, the most nearby anchor nodes are provided.)

### 3.5. Position Calculation

The most likely spatial position of the mobile tag is calculated by combining multiple distance estimations. Ranging messages are received both by the mobile tag, as well as on a back-end server. As such, the position can be calculated centralized (e.g., on a remote server) or locally on the tag using a localization algorithm (e.g., particle filter [28] or Kalman filter [29]).

For the position calculation, we implemented a particle filter which is described in Section 4 of [30]. The particle filter uses a set of *N* particles that represents different possible states. The states of each particle are updated with each incoming range, resulting in a position update for every reporting message. The particle server requires incoming ranges from at least three different anchors for a good 2D position estimate, and at least four different anchors on different heights for a 3D estimate.

### 3.6. Further Optimizations

Finally, multiple optimizations are possible to further decrease the number of packets and hence increase the update frequency of the system.

#### 3.6.1. Optimization I: Single Final UWB Message

The first optimization reduces the number of messages transmitted by using only one single ranging-final message (Figure 5a). This reduces the number of slots from 2+3n to 3+2n. Because the range is calculated on all anchors at the same time, the reporting is done at the end of the superframe.

#### 3.6.2. Optimization II: Multiple Ranging Sequences

The second optimization reduces the amount of sub-GHz synchronization beacons by introducing multiple ranging sequences with the same anchor nodes within a superframe. All slots of the superframe—except the sync slot—is repeated several times (Figure 5b). This is only possible if the selected set of anchors does not change during the superframe duration. Similarly, the long-range sub-GHz reporting messages can also be combined, reducing the number of report slots, but resulting in longer duration between range reports. The number of slots is reduced from k(3+2n) to 1+n+k(2+n), where k is the number of sequences.

#### 3.6.3. Optimization III: Concurrent Radios

Since the system relies on two separate radio chips, it is possible to further reduce the superframe duration (and hence increase the update frequency) by assuming that the sub-GHz radio can transmit the range reports in parallel with the UWB transmissions using a CSMA/CDMA protocol (Figure 5c). Therefore, the superframe no longer requires dedicated reporting slots. This optimization reduces the number of required localization slots to 1+k(2+n).

## 4. Implementation and Evaluation Setup

The localization system was implemented using a custom designed hardware board and a Contiki based software implementation. In Table 2 a full description of the hardware and software can be found. Both the hardware and software design are available open source [14].

### 4.1. Hardware Board

Since no current UWB hardware boards support sub-GHZ technologies, an existing sub-GHz IoT platform was used: the Zolertia Re-Mote [31]. The Zolertia Re-Mote supports two low-power IoT radios (CC1200 and CC2538), but only the CC1200 sub-GHz radio was used in this paper. For this IoT platform, a custom UWB add-on board using the Decawave DW1000 [32] UWB Transceiver was designed. To improve the performance, the UWB hardware board supports an SMA connector, allowing to utilize external antennas instead of chip antennas.

### 4.2. Software Implementation

The Zolertia Re-Mote supports the Contiki OS which includes a full IoT protocol stack for wireless mesh backbones using open IPv6 supported standards such as RPL and 6lowpan. To implement the MAC layer, a custom implementation was required. To this end, the multi-technology MAC protocol was implemented using the Time Annotated Instruction Set Computer (TAISC) framework [33], which supports the design of TDMA MAC protocols that use multiple radios.

Since easy installation is one of the main design goals, we also support easy configuration of the system. High level settings of the MAC protocol (e.g., ranging scheme configuration, slot size) and the anchor selection (e.g., number of anchors to range with, anchor selection approach) and low-level settings on the PHY (e.g., bitrate, preamble bits) are run-time (re)configurable from a central server using the Constrained Open Application Protocol (CoAP) [34] open standards for resource reconfigurations. To report the ranging results to the Localization Engine (LE), the MQTT protocol is used.

### 4.3. Radio Settings

Both the sub-GHz radio (CC1200) and the UWB radio (DW1000) support multiple modulation types, which influence several properties of the communication performance such as packet duration, range, scalability, reliability and energy consumption. To implement the MAC protocol, multiple radio settings were evaluated as shown in Table 3.
For the *UWB radio*, two configuration options will be evaluated in the following section. The first setting UWB${}_{slow}$ corresponds to the slowest bitrate and largest preamble. This settings has the highest accuracy ranging, but negatively impacts the update rate due to the longer packet durations. In contrast, the UWB${}_{fast}$ settings result in lowest energy consumption and larger update rates, but result in reduced ranging accuracy. The impact of these settings will be evaluated in the next section.For the *sub-GHz radio*, two different configurations are used during each MAC superframe. (i) The sub-GHz${}_{slow}$ is used by the anchor nodes to report the ranging results to a remote server. This setting uses a low datarate and large preamble settings to maximize the range. (ii) In contrast, the sub-GHz${}_{fast}$ setting uses a higher bitrate and lower preamble to maximize the datarate. This last setting is used by the mobile tag to wake up and synchronize nearby anchor nodes. In this case, high datarates are preferable since the higher datarate reduces the length of the beacon slot of the TDMA, and the reduced range has no impact since the tag only needs to wake up anchor nodes which are nearby.

### 4.4. Proof-of-Concept Deployments

Finally, the designed UWB system has been experimentally validated using two proof-of-concept implementations.
A small-scale set-up of the localization system has been used for autonomous drone navigation for inventory-taking (Figure 6a). The drone is localized using battery-powered anchor nodes and autonomously traverses a path next to warehouse racks. All assets are equipped with QR codes, which are scanned with a drone-mounted camera. A video of the demo can be found on https://youtu.be/NyL4-R8QEH8.A large-scale installation of the system has been installed for tracking runners during their training in the sporting hall in Ghent (Figure 6b). The indoor training track is located beneath the main tribune, and as such is shielded from the outside by metal and reinforced concrete. No existing infrastructure (Wi-Fi, DC power, Ethernet or 4 G) is available. The designed infrastructure-light UWB localization solution was hence a good candidate to provide training insights based on location tracking. The whole 400 m circular track was covered using 20 anchor nodes, all of which could communicate with a single gateway with Ethernet uplink due to the use of long-range sub-GHz communication.

## 5. System Performance Evaluation

Finally, the performance of the MAC design is evaluated both theoretically and experimentally. The system is analyzed in terms of update frequency, current consumption, accuracy and infrastructure requirements (communication range).

### 5.1. Update Frequency

Most localization approaches (Kalman filter, particle filters) can provide a new position update every time a ranging message is received. As such, the update frequency is defined as the average number of ranging messages that are received over time. To calculate the ranging updates per time unit, the packet durations should be known.

The transmission time of a UWB packet depends on the payload size and the selected physical settings. The frame starts with $b_{pre}$ bits in the preamble, following by the SFD (Start of Frame Delimiter) bits which indicates the start of the frame. The number of SFD bits $b_{SFD}$ is 8 when the bitrate is set to 110 kbps and 64 in the other cases. The frame consist of the physical header with $b_{header}=21$ bits, the payload with $b_{payload}$ bits and several bits for Reed Solomon error correction.

The duration for each transmitted symbol is different in the preamble, the header and the actual payload. It depends on the chosen PRF for the preamble symbol and the bitrate for the header and payload symbols. The exact symbol timings can be found in the DW1000 datasheet [32] and are denoted as $\tau_{SHR}$, $\tau_{PHR}$ and $\tau_{DATA}$. Note that all variables required for calculation are defined in Table 4.

To calculate the packet duration for the UWB messages, the following formulas are used:(7)$$\begin{array}{ccc} T_{sync} & = & \left(b_{pre}+b_{SFD}\right)\times \tau_{SHR} \end{array}$$
(8)$$\begin{array}{ccc} T_{header} & = & b_{header}\times \tau_{PHR} \end{array}$$
(9)$$\begin{array}{ccc} T_{data} & = & \left(b_{payload}+b_{rs}\right)\times \tau_{DATA} \end{array}$$
(10)$$\begin{array}{ccc} T_{tx} & = & T_{sync}+T_{header}+T_{data} \end{array}$$

The transmission time of sub-GHz packets depends on the modulation type (2-GFSK or 4-GFSK) and the bitrate (1.2 kbps to 500 kbps). The preamble and synchronization bits are transmitted using 2-GFSK while the remainder of the packet can be sent using 2-GFSK or 4-GFSK. In Table 5, the packet durations are shown for the slowest and fastest configuration for each UWB and sub-GHz. The packet duration for the synchronization message is calculated assuming a payload of 60 Bytes, which makes it possible to schedule up to 20 anchors in one superframe.

Using these different transmission durations, we can define the lengths of the different slots using Equation (Equation 11). The slot length is the sum of the packet transmission duration, slot margins and time for hardware dependent properties such as radio turnover time and micro-controller instruction delay. To calculate the ranging update frequency $f_{update}$, we first need to determine the superframe duration, which is simply the sum of the duration all slots in the superframe (Equation (12)). The update frequency is now found by calculating the number of superframes that fit in one second, multiplied with the number of anchors *n* inside one superframe (Equation (13)).
(11)$$\begin{array}{ccc} T_{slot} & = & T_{tx}+t_{radio}+t_{instructions}+t_{margin} \end{array}$$
(12)$$\begin{array}{ccc} T_{superframe} & = & \sum_{s\in S}T_{slot}\left(s\right) \end{array}$$
(13)$$\begin{array}{ccc} f_{update} & = & nT_{superframe}^{-1} \end{array}$$
where *S* is the collection of all slots in the superframe and *n* is the number of selected anchors in the superframe.

#### 5.1.1. Theoretical Calculation

The theoretical update frequency is calculated ignoring any delays caused by SPI bus, instruction delay and radio transition times $\left(t_{radio}=t_{instructions}=t_{margin}=0\right)$. The corresponding update frequencies are shown in Table 6. For the basic protocol, the update rate varies between 125 and 790 Hz depending on the PHY settings. After applying the discussed optimizations from Section 3, the update rate varies between 343 and 2892 Hz depending on the PHY settings.

#### 5.1.2. Experimental Validation

For the validation of the proposed system, three different slot sizes are used (see Table 7). (i) The UWB slot is used for all UWB communication and includes the time of the transmission of a ranging-final message together with all device specific delays for instruction, SPI-communication and radio turnover times. (ii) A second slot is used for the synchronization, and contains a payload of 60 bytes, which includes a header of 12 Bytes, 8 Bytes for synchronization and room for ranging with 20 anchors in one superframe. (iii) The final slot is used for reporting and requires only 22 Bytes. The instruction delays are hardware dependent and are measured using a logic analyzer. The experimental measured update frequency for the different MAC implementations are shown in Table 6. These results are hardware dependent and can be further improved, e.g., by using DMA (Direct Memory Access) to further optimize instruction timings.

### 5.2. Current Consumption

Because UWB is known to be very energy consuming, it is advantageously to disable the UWB radio as much as possible, instead relying on the sub-GHz radio to wake the anchor nodes when needed.
The current consumption for transmission and reception on the UWB radio are depending on several PHY settings. Higher frequency channels consume more power than lower frequency channels. Long preamble, payload length and high datarate results in longer packet duration and thus more current demand. The exact current consumption in TX and RX can be found in the Decawave DW1000 user manual [32], and varies between 48 and 117 mA. In contrast, the UWB radio-chip consumes only 1 μA in SLEEP-mode and only 100 nA in DEEPSLEEP (the chip can be woken up by setting a programmed sleep count or by using an external pin. This requires only the low power oscillator and the internal sleep counter to be active. To enable external micro-controller access, the lowest power state is the INIT-state, which consumes 4 mA. For full speed SPI access and internal clock, the IDLE-mode consumes 12 mA.)Also the current consumption of the CC1200 depends on the chosen output power (up to 16 dB) and the PHY settings of the radio. The consumption for the chip for reception varies from 17 to 23 mA and for transmission from 34 to 45 mA [35]. In SLEEP mode, it only consumes 0.5 μA.

The anchors can set their UWB radio in SLEEP-mode while they are not selected by the tag. Therefore, they only listen on the beginning of the superframe for the synchronization message using the sub-GHz radio. If we compare this to most UWB solutions, this is a significant reduction in current consumption, since these are constantly listening for preambles (which is the most energy consuming operation of UWB) of possible UWB-messages. The same is true for anchors which are selected to range with. These will only power on the UWB radio on the slots they are supposed to range on. The current consumption on the tag is much higher, since the tag is constantly transmitting or receiving to range with the anchors.

The resulting current consumption (when more interested in the required power, the given current can be multiplied with the required voltage, which is 3.3 V) of the UWB radio and sub-GHz for the most energy consuming configuration (e.g., slowest setting) is shown in Table 8. Most commercial and academic anchor nodes continuously consume on average about 130 mA. In contrast, our anchor nodes consume only 3.4 mA when they are stand-by, and 26.6 mA when they are selected for ranging.

### 5.3. Localization Accuracy

The accuracy of the proposed system is tested in lab environment for small distances (between 1 and 5 m) where the ground-truth is determined with a mm-accuracy laser meter. For longer distances (between 50 and 75 m), the accuracy was tested on a empty parking lot where a roll-meter and reference lines on the ground were used for determining the ground truth. The distribution of these samples is given in Figure 7, together with a cumulative distribution function of the error of the measurements. The mean reported distances are 49.99 and 74.95 m, deviating only 1 and 5 cm from the real distance.

### 5.4. Maximum Communication Range

The maximum communication range determines how many anchor nodes should be installed, which is crucial for cost-efficient deployments in large warehouses. The maximum communication and maximum interference range is defined by the threshold value for distance or sensitivity power level after which the packet cannot be decoded or detected anymore. The sensitivity level can vary when the modulation setting is changed [32], and the distance threshold can increase when increasing tx transmission power level. The signal power that arrives at the receiver has suffered from path loss can be expressed Friis’ path loss formula:(14)$$P_{r}\left[dBm\right]=P_{t}\left[dBm\right]+G\left[dB\right]-L\left[dB\right]-20\log_{10}\left(\frac{4\pi f_{c}d}{c}\right)$$
where $P_{r}$ and $P_{T}$ are the received and the transmitted power, *G* includes the antenna gains, L includes any PCB losses in the system, $f_{c}$ is the center frequency of the channel and *d* is the distance in meters between the transmitter and the receiver.

Because typical radio waves are reflected and obstructed by all objects illuminated by the transmitter antenna, the signal is also exposed to the two ray ground model. The total received energy can be modeled as the vector sum of the direct transmitted wave and one ground reflected wave. The two waves are added constructively or destructively depending on their phase difference at the receiver. The magnitude and phase of the direct transmitted wave varies with distance traveled. The magnitude of the reflected wave depends on total traveled distance and the reflection coefficient.

Figure 8a shows the path loss for the UWB radio calculated both theoretically (green line) and validated experimentally (red dots). Our current UWB board with omni-directional antenna allows reliable ranging up to 80 m, and occasionally (disregarding the Fresnel zone) up to 220 m. Figure 8b shows that the theoretical range of the sub-GHz radio goes up to several kilometers. As such, it is possible to install large warehouses without requiring excessive number of anchor nodes or cabling.

## 6. Future Work

Currently, the set of anchor nodes is selected statically based on the current drone position, whereby each time the nearest anchor nodes are selected. In environments with many obstacles, the nearest anchor nodes are not always the most reliable due to fading, reflections, absorption, etc. In complex environments, the mobile tag can take into account the quality of the information obtained from anchor nodes. For example, the Channel impulse response (CIR) could be used by a machine learning algorithm to obtain error estimates and to detect line-of-sight information, allowing to dynamically update the set of selected anchors nodes for the next superframe.

The correct placement and calibration of the anchors is a crucial step in the installation process to get accurate results. One can manually place each anchor and measure the difference to a reference point with a laser meter, or using reference points on the building-scheme to measure distance. However, this can be a lot of work which is not guaranteed to be error-free. Therefore, auto calibration of the anchors can determine their relative position without any installation process. In fact, anchor calibration is an interesting research track which requires a paper on its own to discuss the topic. Many approaches are already documented in literature, e.g., using multilateration, Linear Programming (LP) or Semidefinite programming (SDP).

## 7. Conclusions

The high accuracy and low cost of UWB radios for indoor positioning makes it the perfect technology for tracking drones in warehouses for autonomous inventory-taking. However, most UWB solutions require communication infrastructure and are energy consuming, thereby limiting placement options and increasing deployment costs. To remedy this, we designed a multi-technology UWB MAC protocol for localization that allows battery- or energy harvesting operated anchor nodes and does not require existing infrastructure such as Wi-Fi or Ethernet. To wake-up anchor nodes, the tag uses low-power sub-GHz messages to activate a list of anchors. Anchor nodes consumes less energy since they only have to listen to synchronization messages, and use TDMA to activate their UWB radio during the specified slot. The overall system potentially has a high update rate, theoretically up to 2892 Hz (without considering hardware dependent properties) and a localization accuracy of 5 cm. Anchor nodes consume only 3.4 mA in stand-by mode. The overall system has been validated using two large-scale set-ups: one for drone tracking in a warehouse mock-up, and one for localizing runners in a challenging infrastructure-less indoor running track.

## Figures and Tables

**Figure 1 sensors-19-00467-f001:**
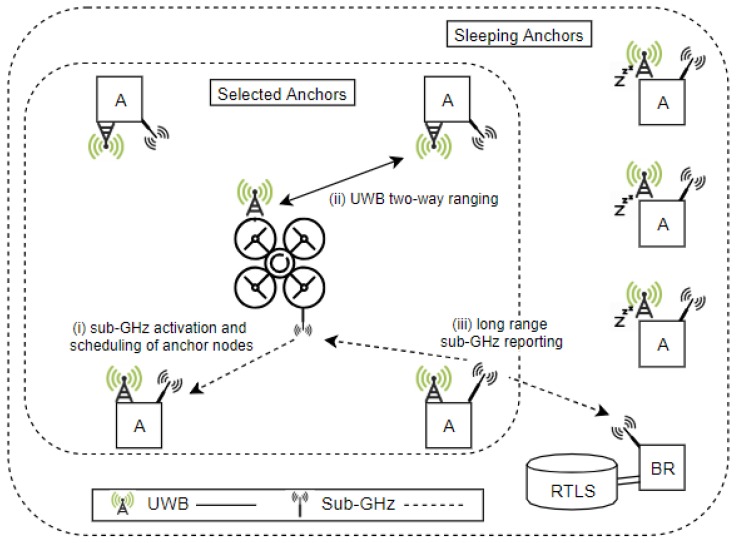
High-level system diagram of the UWB localisation system with battery powered anchor nodes.

**Figure 2 sensors-19-00467-f002:**
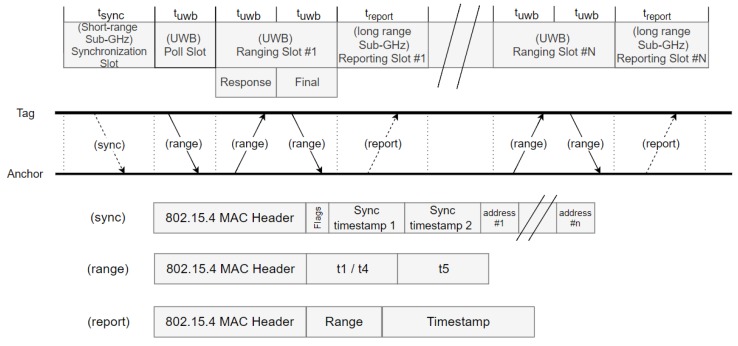
(**Top**): MAC superframe structure with different slots and messages for synchronization, ranging and reporting. Sub-GHz messages are indicated using dotted lines, UWB messages using full lines. (**Bottom**): frame formats. The range message contains two fields for timestamps. The first one is used only in the ranging poll and final message. The second one is only used in the final message.

**Figure 3 sensors-19-00467-f003:**
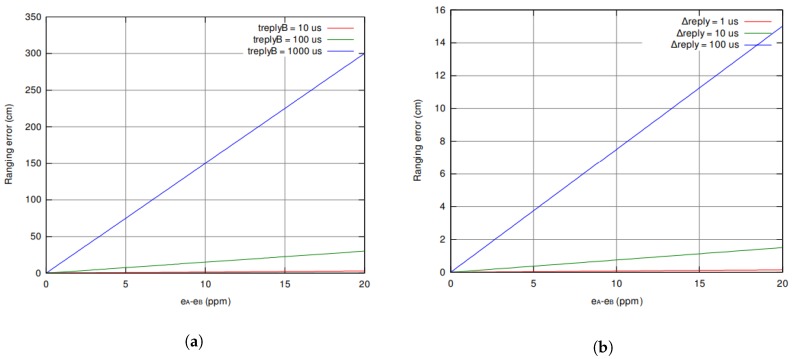
The ranging error due to clock drift in TWR scheme (**a**) is increasing faster than when using SDS-TWR scheme (**b**) [25].

**Figure 4 sensors-19-00467-f004:**
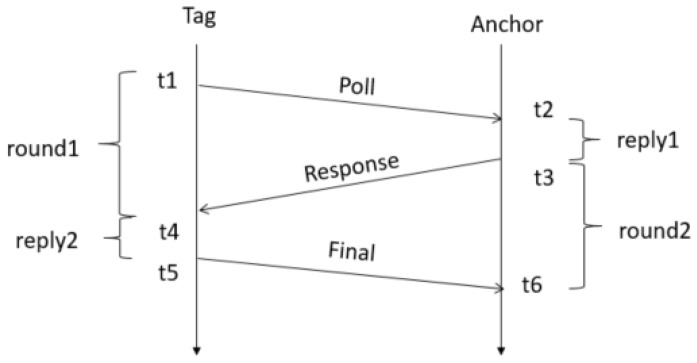
Three-way message exchange using SDS-TWR ranging scheme

**Figure 5 sensors-19-00467-f005:**
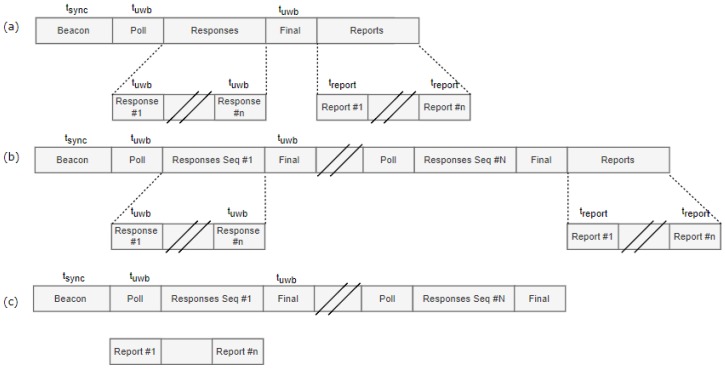
MAC protocol optimizations. (**a**) Optimization 1: Using only a single final UWB message. (**b**) Optimization 2: using multiple ranging sequences within a superframe. (**c**) Optimization 3: using the sub-GHz and UWB radio in parallel.

**Figure 6 sensors-19-00467-f006:**
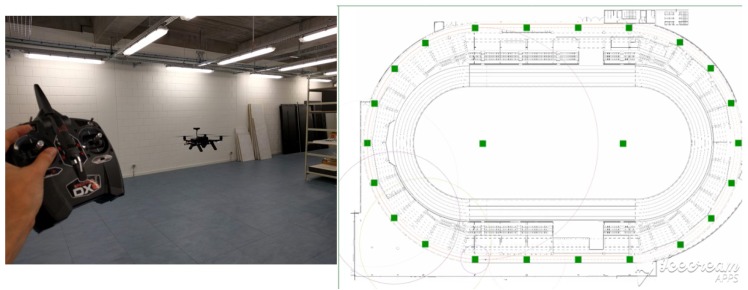
Validation of system. On the left image the drone is using the UWB system to perform an autonomous flight for inventory-taking. In the right image the UWB system is used for tracking a runner on a running track.

**Figure 7 sensors-19-00467-f007:**
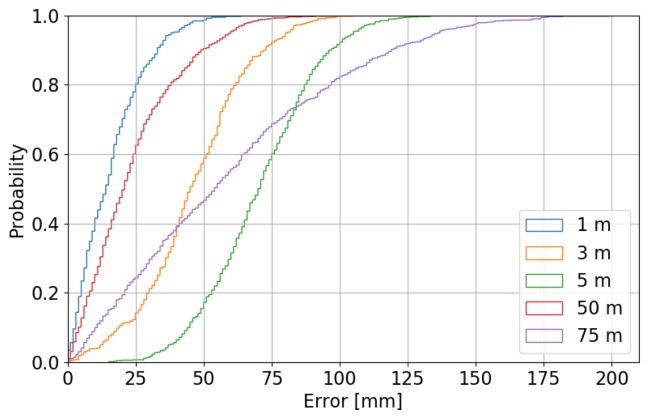
Cumulative distribution function of the ranging errors for different ranging distances.

**Figure 8 sensors-19-00467-f008:**
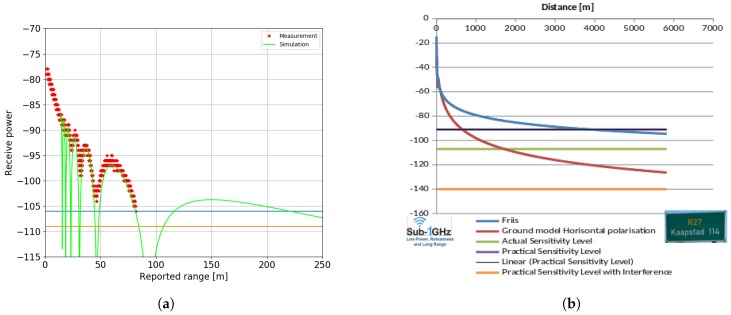
Path loss model for both radio technologies. (**a**) Experimental and simulation-based path loss model for our UWB board. (**b**) Simulated path loss model for the sub-GHz radio [21].

**Table 1 sensors-19-00467-t001:** Overview of infrastructure requirements of existing UWB indoor Positioning solutions.

	Infrastructure	Anchor Powering	Release
[17] Sewio	Wi-Fi/Ethernet	PoE	commercial
[18] eliko	Wi-Fi/Ethernet	PoE	commercial
[19] Tracktio	Ethernet	DC Power	commercial
[20] openRTLS	Zigbee/Ethernet	PoE/DC Power	commercial
[12]	Wi-Fi	DC Power	academic
[13]	Ethernet	DC Power	academic
**Our work**	None	Battery / energy harvester	academic

**Table 2 sensors-19-00467-t002:** UWB localization system specifications.

System on Chip	ARM Cortex-M3
Clock speed	32 MHz
Memory	32 KB RAM 512 KB FLASH
Radios	CC2538: IEEE 802.15.4 (2.4 GHz) CC1200: sub-GHz IEEE 802.15.4 g (868 MHz) DW1000: UWB IEEE 802.15.4a (3.5–6 GHz)
Antenna	On-board (CC1200 and CC3538) SMA Connector (DW1000)
Protocol stack	Contiki OS
MAC Design Framework	Time Annotated Instruction Set Computer (TAISC) [33]

**Table 3 sensors-19-00467-t003:** Modulation coding schemes for UWB and sub-GHz.

	**Channel**	**Datarate**	**Preamble**	**PRF**
$UWB_{slow}$	Ch1	110 kbps	2048	16 MHz
$UWB_{fast}$	Ch5	6.81 Mbps	256	64 MHz
	**Modulation**	**Datarate**	**Preamble**	**Sync**
sub-$GHz_{slow}$	2-GFSK	50 kbps	4 Bytes	2 Bytes
sub-$GHz_{fast}$	4-GFSK	1 Mbps	24 Bytes	2 Bytes

**Table 4 sensors-19-00467-t004:** Variable definition.

***n***	number of anchors scheduled in superframe
$b_{pre}$	number of Preamble bits
$b_{SFD}$	number of Start Of Frame bits
$b_{rs}$	number of Reed Solomon bits
$b_{payload}$	number of Payload bits
$\tau_{SHR}$	synchronization symbol duration
$\tau_{PHR}$	header symbol duration
$\tau_{DATA}$	data symbol duration
$t_{instruction}$	instruction delay
$t_{radio}$	radiochip delay
$t_{margin}$	timeslot margin

**Table 5 sensors-19-00467-t005:** Packet duration for different MCS and message types.

	$T_{\mathrm{poll}}$ (UWB)	$T_{\mathrm{reply}}$ (UWB)	$T_{\mathrm{final}}$ (UWB)	$T_{\mathrm{sync}}$ (sub-GHz)	$T_{\mathrm{report}}$ (sub-GHz)
Slowest setting	2930 μs	2560 μs	3310 μs	3840 μs	1180 μs
Fastest setting	310 μs	300 μs	320 μs	900 μs	600 μs

**Table 6 sensors-19-00467-t006:** Localization update frequency for the UWB localization system, both theoretically (ignoring hardware dependent delays) and experimentally.

	Theoretic Update Frequency (Hz)		Experimental Update Frequency (Hz)	
	Min.	Max.	Min.	Max.
Basic	125	790	65	127
Optimization I	206	1035	98	178
Optimization II	284	1841	136	276
Optimization III	343	2892	166	372

**Table 7 sensors-19-00467-t007:** Slot sizes for experimental update frequency.

	${\mathrm{slot}}_{\mathrm{uwb}}$	${\mathrm{slot}}_{\mathrm{sync}}$	${\mathrm{slot}}_{\mathrm{report}}$
Slowest setting	5400 μs	6800 μs	4000 μs
Fastest setting	2400 μs	3800 μs	2800 μs

**Table 8 sensors-19-00467-t008:** Current consumption of the anchor nodes in different power states with different MAC implementations. The last line shows the total current consumption for the complete superframe.

Power State	$I_{\mathrm{avg}}$ [mA]	Stand-by Anchor (No Slot)		Active Anchor (One Slot)		Always-RX Anchor (Traditional Approach)	
		%	mA	%	mA	%	mA
UWB RX	133	-	-	10.3	13.699	89.1	118.51
UWB TX	102	-	-	5	5.1	10.9	11.12
UWB SLEEP	0.001	100	0.001	84.7	0.001	-	-
SubGHz RX	23	14.8	3.404	14.9	3.427	-	-
SubGHz TX	45	-	-	9.8	4.41	-	-
SubGHz SLEEP	0.005	85.2	0.004	75.3	0.004	-	-
**Superframe**		100	**3.409**	100	**26.641**	100	**129.63**

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
