# Peer review of "UWB Localization with Battery-Powered Wireless Backbone for Drone-Based Inventory Management"

_sensors, 2019, doi:10.3390/s19030467_

Round 1
Reviewer 1 Report
This paper proposed a UWB based localization solution for drone-based inventory management. To ease deployment, battery powered wireless backbone is combined with the UWB localization by TDMA-based MAC protocol. The proposed solution is analyzed in theory and validated by experiments in real life. As a practitioner as well as researcher in this area, the reviewer is attracted by the motivation and experimental validated results in this paper. The overall presentation of the paper is good too.
I recommend minor revision before final acceptance as the authors need to address the following concerns:
1, in section 3.1: I hope to see a system level diagram of proposed solution, showing the key elements and operations of each elements to help readers to follow the text more easily.
2, in the equation 1-4, the two way ranging scheme applied is not novel, so proper citation, e.g. the 802.15.4a standard, is a must to avoid plagiarism. Citation in the caption of Figure 3 is not enough.
3, in section 3.4, the selection of anchors is a very tricky part according to the reviewer’s practical experience. I hope to see more modest description of the solution used in this paper, and more realistic/critical discussion about ML based solution (I will be surprised if it is mature enough).
4, in table 5, the deviation between theoretical and experimental results is quite big. I what to see more explanation, e.g. is it only due to the radio turnover time? Or any other important factors ignored? It is better to modify the equation 7-11 to give a more realistic theoretical estimation.
5, more importantly, the authors must moderate the claim of 2892Hz update rate in the abstract and anywhere else, since it is an optimistic number based on rough theoretical estimation which is obviously not achievable in experiments. I was surprised at the first glance when I receive the review invitation. To be honest, it was this number that attracted me to review this article.
6, before giving the conclusion, more discussions on limitations are needed:
1) There seems no slot reserved for packet retransmission of UWB or sub-GHz. What is the consideration of the authors? How about the performance when there is packet loss? Note, the authors have stated warehouse environment is challenging for radios, which is fully agreed by the reviewer. The reviewer believes the update rate should be noticeably reduced.
2) The scalability of the RPL used in this work is questionable. When the number of hops increases, the latency and reliability of the network becomes bad according to the reviewers experiences. I am not fully convinced about the benefit of scalability claimed by the authors. Either give strong evidence to keep the current statements or soften the statements.
3) The experiments are done in quite “comfortable” environment instead of real warehouses. More critical and modest discussion about the performance in real warehouse is needed.
4) Another important challenge when deploy UWB is the positioning and calibration of the anchors. The authors are encouraged to discuss their considerations. Reference is :
[7] “Location aided commissioning of building automation devices enabled by high accuracy indoor positioning”, Journal of Industrial Information Integration, 2017, https://doi.org/10.1016/j.jii.2017.12.002
Author Response
see coverlettter in attachment

Reviewer 2 Report
The paper is generally well written and has merits. It contains many practical suggestions which can be helpful to other researchers or even more probably to designers of indoor localization/navigation systems based on wireless technology (UWB and subTHz transceivers). The propositions of using both UWB and subTHz modules to avoid constant operation of energy consuming UWB modules and various protocol optimizations are original and interesting.
The paper however has some drawbacks and could be further improved. It is more technical than scientific in nature. Many important aspects of the system, e.g. description of the particle filter used for the state estimation is only mentioned.
1. I would suggest expanding the 3.5 section of the paper.
2. The section 6 which describes the testing scenarios is later than the results obtained in these scenarios. Typically, the methodology of testing precedes the test results.
3. The equation (1) should be corrected, as we subtract tau_reply from tau_round and not vice versa.
4. The power consumption in section 5.2 should not be given in mA but in watts.
5. Many terms and abbreviations should be explained when used for the first time, e.g. “Industry 4.0” (line 14), “X-as-a-service” (line 16), PoE (line 101), RPL (line 115), RTLS (line 148), TWR (line 156).
6. There are some minor mistakes thus the paper requires check, e.g. “extend” instead of “extent” (line 92), “strong” instead of “strongly” (line 156), “onyl” instead of “only” in Fig. 4 caption, “the be” instead of “to be” (line 316).
7. The labels and legend in Fig. 3 are too small. In lines 358 and 361 there are no numbers of figures but “??”.
Author Response
see coverletter in attachment
